# Directly Optimizing Natural Language Explanations for Behavioral Faithfulness: Simulatability and Recoverability

Advaith Malladi [1]   Shashank Srivastava [2]

## Abstract

Natural-language explanations are widely used to interpret machine learning models, yet many prioritize human plausibility over accurately reflecting or predicting model behavior. Prior approaches often rely on human-written rationales, producing post-hoc explanations that neither align with the model's decision function nor generalize. We introduce OPEX , a natural-language explanation model that directly optimizes for *behavioral faithfulness*: the ability of an explanation to reflect and predict a model's observable input–output behavior. OPEX is trained using reinforcement learning with Group Relative Policy Optimization (GRPO), optimizing two complementary metrics: *recoverability*, which measures whether explanations recover model predictions on seen examples, and *simulatability*, which measures prediction of model behavior on unseen inputs. Across structured and text-based tasks, OPEX achieves high simulatability ($\sim$0.85) and recoverability ($\sim$0.99), outperforming `GPT-4o`, `LLaMA-3.3-70B`, MaNtLE, Chain-of-Thought (CoT)-based models, and human-written explanations, despite using an 8B-parameter backbone. Human user studies show a 15% improvement in classification accuracy over competent baselines. Link to Code and OPEX weights: ⬤ github.com/advaithmall/OPeX

## 1. Introduction

Natural-language explanations have become a common interface between machine learning models and their users (Lakkaraju et al., 2022). They are used to justify predictions, support decision-making, and help practitioners anticipate

[1] IIIT Hyderabad [2] UNC Chapel Hill. Correspondence to: Advaith Malladi <advaithmalladi02@gmail.com>.

*Proceedings of the 43rd International Conference on Machine Learning*, Seoul, South Korea. PMLR 306, 2026. Copyright 2026 by the author(s).

model behavior. Yet many explanations that appear convincing to humans fail at a more basic task: they do not reliably reflect or predict the behavior of the model they are meant to explain. Instead, they often function as post-hoc narratives that are plausible and linguistically fluent, but weakly coupled to the model's actual decision process (Atanasova et al., 2023).

A central challenge in explainability is *faithfulness* (Jacovi & Goldberg, 2020): explanations should reflect how the model actually behaves, not just sound plausible. In this work, we focus on *behavioral faithfulness*: whether an explanation enables accurate reproduction of a model's observable input–output behavior. This contrasts with *mechanistic faithfulness*, which targets correspondence to internal representations or parameters. While mechanistic faithfulness is valuable, it is often inaccessible to end users and infeasible for black-box or proprietary models. Behavioral faithfulness is instead aligned with how explanations are used in practice: to anticipate and reason about model predictions.

We operationalize behavioral faithfulness with two metrics. *Recoverability* measures whether an explanation can recover the model's predictions on the examples used to generate it, capturing fidelity under interpolation. *Simulatability* asks whether the explanation enables prediction of model behavior on previously unseen inputs, capturing generalization beyond seen examples. Together, they assess whether an explanation is a reliable surrogate for the model's decision function. Existing interpretability methods fail these tests. Feature- and rule-based approaches like LIME and Anchors (Ribeiro et al., 2016; 2018) highlight influential inputs but rarely yield semantic explanations users can apply to new cases. Natural-language explanation systems, such as MaNtLE (Menon et al., 2023) and WT5 (Narang et al., 2020), are trained on human-written rationales, prioritizing plausibility and producing articulate but weakly predictive explanations. Their reliance on human annotations also limits scalability and makes domain adaptation costly. Recent works also explore Chain-of-Thought as interpretable explanations (Wei et al., 2023); however, they sometimes fail to represent the model's actual decision process (Barez et al., 2025).

To address these challenges, we introduce OPEX (Optimized Policy for Explanations), a natural-language ex-

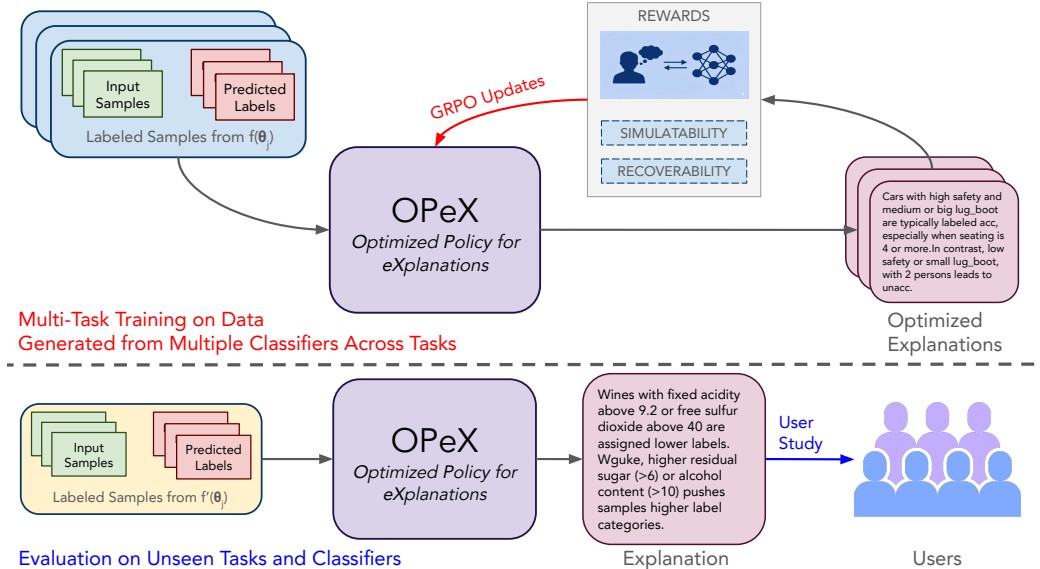

*Figure 1.* OPEX is a natural-language explainer that produces faithful and simulatable explanations of a model's decision rationale. It is trained via multi-task learning across data generated from multiple classifiers and tasks, using GRPO with Simulatability and Recoverability as reward signals. OPEX generalizes to unseen tasks and classifiers in zero-shot settings. Human user studies and automated evaluations show that OPEX explanations enable more accurate prediction of model behavior compared to competing explanation frameworks.

planation generation system designed to align explanations directly with classifier behavior rather than human intuition (Figure 1). Instead of learning to imitate human rationales, OPEX learns explanations that maximize behavioral faithfulness. We frame explanation generation as a reinforcement learning problem and train OPEX using Group Relative Policy Optimization (GRPO). Explanations are optimized directly against reward functions for simulatability and recoverability (§3.3), eliminating the need for human-written reference explanations. Following recent work that uses LLMs as proxies for human evaluators (Poché et al., 2025; Bona et al., 2024), we employ a User-proxy LLM to compute reward signals during training (§3.3). To guard against evaluator leakage and stylistic reward hacking (Xu et al., 2024; Cheng et al., 2025), we include synthetic datasets whose labeling rules are unknown to the evaluator, ensuring that high recoverability and simulatability reflect genuine learning of model behavior.

Across a diverse set of structured and text-based datasets, OPEX explanations are consistently more behaviorally faithful than existing baselines, improving simulatability by up to 18% and recoverability by up to 28%. Human user studies show that OPEX explanations enable users to predict model labels with 15% higher accuracy than competing explanation methods. Our contributions are: (1) we formalize *behavioral faithfulness* for natural-language explanations using two complementary metrics: *recoverability* and *simulatability*; (2) we introduce OPEX , a GRPO-trained explanation model that directly optimizes these metrics with-

out relying on human-written rationales; and (3) we show that OPEX produces significantly more faithful explanations than strong LLM baselines, including `GPT-4o` and `LLaMA-70B`, MaNtLE, and Chain-of-Thought based approaches across tasks, classifiers, and human user evaluations.

## 2. Related Work

**Instance-level and feature-based explanations.** A large body of work focuses on explaining individual model predictions through local approximations or feature attributions. LIME (Ribeiro et al., 2016) fits sparse linear surrogates around a target instance to identify influential features, while Anchors (Ribeiro et al., 2018) extracts high-precision, model-agnostic rules that capture sufficient conditions for a prediction. Shapley-value–based methods (Lundberg & Lee, 2017) provide theoretically grounded feature attributions by decomposing predictions into additive contributions. While effective for highlighting local sensitivities, these approaches are inherently instance-specific and do not yield reusable or generalizable descriptions of model behavior, nor do they produce explanations in natural language.

**Natural-language explanations and rationales.** Subsequent work explores generating explanations directly in natural language. MaNtLE (Menon et al., 2023) produces textual explanations using a model-agnostic explainer trained on human-written rationales, optimizing for agreement with human explanations rather than alignment with the target

model's decision function. Related approaches generate free-form rationales alongside predictions, including CAGE (Rajani et al., 2019) and WT5 (Narang et al., 2020). While these methods produce fluent explanations, they remain tied to individual inputs and rely heavily on human annotations, limiting scalability and generalization. More recent work examines self-generated explanations from large language models (Huang et al., 2023; Dehghanighobadi et al., 2025; Turpin et al., 2023), finding that such explanations are often post-hoc, instance-level, and weakly coupled to the model's underlying decision behavior.

**Learning frameworks for explanation alignment.** Beyond explanation architectures, several lines of work motivate the training paradigm adopted here. Multi-task learning improves data efficiency and generalization by sharing structure across tasks (Yu et al., 2024), supporting our multi-task formulation across structured and text-based domains. Reinforcement Learning from Human Feedback (RLHF) (Ouyang et al., 2022; Stiennon et al., 2022) provides a general framework for aligning language models with desired behaviors but commonly relies on on-policy optimization methods such as PPO. Recent surveys highlight limitations of these approaches and motivate alternative optimization strategies (Wang et al., 2026). Group Relative Policy Optimization (GRPO) (Shao et al., 2024) removes the need for a learned critic by optimizing relative preferences among sampled outputs, offering improved stability and efficiency for large-scale language model training.

**Positioning of this work.** We diverge from previous work in focusing explicitly on behavioral faithfulness: whether an explanation enables accurate reproduction and prediction of a model's observable input–output behavior. Rather than imitating human-written rationales or generating instance-level justifications, we directly optimize natural-language explanations for simulatability and recoverability using reinforcement learning. This formulation enables explanation learning without human annotations, supports both structured and text-based tasks, and yields explanations that function as reusable, testable surrogates for model behavior.

## 3. OPEX

This section formalizes the explanation learning problem, introduces the OPEX architecture and its variants, and describes the metrics and training procedure used to optimize explanations for behavioral faithfulness.

### 3.1. Problem Statement

We consider a fixed classifier $f : \mathcal{X} \rightarrow \mathcal{Y}$, where $\mathcal{X}$ denotes the input space and $\mathcal{Y}$ the label space. Given a set of input–label pairs $\{(X_i, Y_i)\}_{i=1}^{N}$ sampled from the behavior of $f$, the goal of OPEX is to produce a natural-language

explanation $e \in \mathcal{E}$ that captures the decision logic of $f$. We define an explanation $e$ as a natural-language description that enables prediction of $f$'s outputs without access to the classifier itself. Concretely, a valid explanation should (i) reproduce the classifier's predictions on the examples used to generate it, and (ii) generalize to accurately predict the classifier's behavior on unseen inputs. In this sense, explanations act as behavioral surrogates for $f$, rather than post-hoc justifications or descriptions of individual samples.

### 3.2. Explanation Model

Most prior approaches frame explanation generation as a supervised natural-language generation problem, typically training on human-written rationales. In contrast, OPEX is trained to align explanations directly with the observable behavior of the target classifier rather than with human annotations. All variants of OPEX are initialized from `LLaMA-3.1-8B-Instruct`, allowing the model to leverage the linguistic competence and generalization capabilities acquired during large-scale pretraining. Given a prompt consisting of $N$ input–label pairs $\{(X_i, Y_i)\}_{i=1}^{N}$, OPEX generates a single natural-language explanation intended to summarize the classifier's decision logic across these examples and generalize beyond them. Input samples are linearized and formatted using a fixed prompt template(Table 9), which instructs the model to describe decision rules in terms of explicit feature conditions and value ranges. Importantly, explanations are generated at the *task level*: each explanation is intended to characterize a classifier's behavior, rather than justify an individual prediction. We introduce two primary variants of OPEX . OPEX-R is optimized for *recoverability*, encouraging explanations that closely reproduce the classifier's behavior on the examples used to generate them. OPEX-S is optimized for *simulatability*, encouraging explanations that generalize to accurately predict classifier behavior on unseen inputs. We also consider a variant optimized for both objectives (§4). OPEX explanations can be found in Table 1

### 3.3. Evaluation Metrics

To assess whether an explanation functions as a faithful behavioral surrogate for a target classifier, we evaluate it using two complementary metrics: recoverability and simulatability. Intuitively, both metrics measure whether a reader can use the explanation to reproduce the classifier's behavior. Recoverability evaluates this on the same examples used to generate the explanation, while simulatability evaluates whether the explanation generalizes to new unseen examples. Together, they capture whether the explanation describes a reusable decision rule rather than a plausible but behaviorally unfaithful rationale.

Both metrics ask the same question at different levels of

| Model | Explanation |
|---|---|
| OPEX | The target is driven by recurring interaction patterns, with buying, maintenance, and safety forming the primary decision boundary. High buying prices almost always lead to unacc, especially when combined with high maintenance or low safety. When buying is low or medium, low maintenance and high safety enable acc, while unfavorable values revert the outcome to unacc. Capacity-related features (doors, persons, lug_boot) act as secondary modifiers, supporting acc only when cost–safety conditions are already favorable. |
| Llama-8B-Instruct | The target depends on several features, including buying, maintenance, safety, number of doors, number of persons, and lug_boot size. Buying and maintenance can be low, medium, high, or very high, while safety can be low, medium, or high. The number of doors, persons, and lug_boot also vary across samples. The target reflects the combination of these feature values. |

*Table 1.* Example explanations generated for the Car Evaluation task. OPEX produces pattern-level explanations that capture feature interactions and decision structure, while Llama-8B-Instruct primarily provides feature descriptions with limited relational insight. Further examples of OPEX explanations can be found in § A.3.

generalization: given only the explanation and an input sample, can a user-proxy correctly predict the classifier's label?

Formally, let $f : \mathcal{X} \to \mathcal{Y}$ denote a target classifier and let $S = \{(X_i, Y_i)\}_{i=1}^N$ be a set of inducing examples sampled from the behavior of $f$, where $Y_i = f(X_i)$. Given $S$, OPEX generates a single task-level natural-language explanation $e$ summarizing the classifier's decision behavior. We then evaluate $e$ with a fixed User-proxy LLM $U$ that receives only the explanation and an input instance, and is asked to predict the classifier's label.

**Recoverability.** Following Menon et al. (2023), recoverability measures how well the explanation supports prediction on the same examples used to elicit it. Specifically, we query $U$ on each inducing example $X_i \in S$ using only $e$, and compute the fraction of correct predictions:

$$\text{Recoverability}(e; S) = \frac{1}{N} \sum_{i=1}^{N} \mathbb{I}[U(e, X_i) = Y_i].$$

Recoverability therefore measures whether the explanation accurately reconstructs the classifier's observed input–output behavior on the examples that were shown to OPEX.

**Simulatability.** Simulatability (Hase & Bansal, 2020) measures whether the explanation generalizes beyond the inducing examples. Let $\hat{S} = \{(\hat{X}_i, \hat{Y}_i)\}_{i=1}^N$ be a disjoint held-out set from the same task and classifier, with $\hat{Y}_i = f(\hat{X}_i)$. We query $U$ on each unseen example $\hat{X}_i$ using only $e$, and compute:

$$\text{Simulatability}(e; \hat{S}) = \frac{1}{N} \sum_{i=1}^{N} \mathbb{I}[U(e, \hat{X}_i) = \hat{Y}_i].$$

Simulatability complements recoverability by evaluating the explanation on unseen examples rather than the inducing examples used to generate it. It therefore measures whether the explanation generalizes to new inputs from the same classifier and task.

**Behavioral faithfulness.** Together, recoverability and simulatability quantify behavioral faithfulness: the extent to which an explanation captures the classifier's observable input–output mapping. Importantly, these metrics do not attempt to recover internal causal mechanisms, latent representations, or mechanistic explanations. Instead, they measure whether the explanation is functionally useful as a predictor of the classifier's behavior.

**User-proxy LLM evaluation.** Following recent work showing that large language models can serve as effective proxies for human evaluators in explainability settings (Poché et al., 2025; Bona et al., 2024), we use a fixed User-proxy LLM to compute both metrics at scale. Given an explanation $e$ generated by OPEX and a sample $X$, the User-proxy LLM is prompted to predict the classifier's label using only $e$ and $X$, without access to the classifier, external knowledge, or ground-truth labels. We report the resulting prediction accuracy on inducing examples as recoverability and on held-out examples as simulatability. Unless otherwise stated, we use `LLaMA-3.3-70B-Instruct` as the User-proxy LLM due to its strong reasoning capability and consistent performance across domains.

### 3.4. Training Procedure

We train OPEX using multi-task learning across a diverse collection of classification tasks that vary in modality, feature dimensionality, and decision structure. This setting reflects the intended use of OPEX as a general-purpose explanation model that must operate across heterogeneous domains without access to task-specific human explanations.

**Training data.** Our training corpus consists of 16 classification tasks: 14 structured (tabular) tasks and 2 text-based tasks. The structured tasks include Abalone, Adult, Census Income, Mushroom, Nursery, Thyroid, MAGIC Gamma Telescope, Wine Quality, Bank Marketing, Online Retail, Occupancy Detection, Air Quality, Rice, and Bike Sharing. These datasets (Kelly et al., 2019) span feature counts from 5 to 22 and exhibit varying degrees of class imbalance and

feature interactions. The text-based tasks are IMDb Reviews (binary sentiment classification) (Maas et al., 2011) and Hate Speech Detection. Human-written explanations are scarce or entirely unavailable for many of these datasets, particularly in structured domains, and even when human rationales exist, they often reflect human intuitions rather than the learned decision rules of a specific classifier. This scarcity and misalignment motivate our choice to learn explanations without human supervision, optimizing directly for behavioral objectives derived from the classifier.

Each training instance is constructed by sampling 25 input–label pairs from a single task and linearizing them into a prompt. The prompt instructs (Table 9) OPEX to infer and articulate the classifier's decision logic based solely on these examples. Explanations are generated at the task level: each explanation summarizes the classifier's behavior across multiple examples rather than justifying a single prediction. The resulting training set contains approximately 8,000 such instances aggregated across all tasks.

**Optimization via reinforcement learning.** Recoverability and simulatability provide noisy, non-differentiable rewards, and no ground-truth explanations. We therefore cast explanation learning as reinforcement learning from feedback and fine-tune OPEX using parameter-efficient LoRA adapters with Group Relative Policy Optimization (GRPO) (Shao et al., 2024). GRPO constructs group-normalized advantages from multiple sampled explanations per prompt, removing the need for a learned value (critic) network and reducing optimization variance. This improves training stability and sample efficiency under sparse, task-specific rewards. During training, OPEX samples multiple candidate explanations per prompt, which are evaluated by a User-proxy LLM to compute recoverability or simulatability rewards, scaled to $[-1, 1]$. These rewards update only the LoRA parameters, keeping the base model frozen and enabling stable multi-task optimization.

**Training variants.** We train two primary variants of OPEX using different reward functions. OPEX-R is optimized for recoverability, encouraging explanations that faithfully reproduce the classifier's behavior on the examples used to generate them. OPEX-S is optimized for simulatability, encouraging explanations that generalize to accurately predict classifier behavior on unseen inputs. Both variants are trained using the same multi-task dataset and optimization hyperparameters. Full training details, prompts, and hyperparameters are provided in § A.

**Training and Evaluation Setup.** OPEX is implemented as a single LLaMA-3.1-8B model with LoRA weights trained via GRPO across many (task, classifier) pairs. Each training prompt is built from $N = 25$ labeled examples drawn from one task–classifier pair using the template in Table 9, and GRPO updates the LoRA weights based on recoverability or simulatability scores assigned by the User-proxy LLM. At inference time, the same model is applied to a held-out (task, classifier) pair without any additional fine-tuning: the general explanation capability is encoded in the LoRA weights, while the particular classifier to be explained is specified entirely by the in-context examples. We evaluate by computing recoverability on the inducing examples and simulatability on held-out test examples, additionally using synthetic tasks to reduce the influence of the User-proxy LLM's pretraining knowledge, and comparing OPEX against prompted LLM baselines throughout.

## 4. Experiments

To evaluate the explanations generated by OPEX , we consider four structured classification tasks, namely Balance Scale, Car Evaluation, Iris, and YouTube Spam, two text-based classification tasks, namely Financial PhraseBank and COVID Tweets Sentiment Analysis, and four synthetic tasks proposed in CLUES (R. Menon et al., 2022). We include synthetic tasks to ensure that the User-proxy LLM does not rely on pre-trained knowledge when classifying samples and to prevent potential knowledge leakage during evaluation.

To assess whether OPEX can explain behavior across a diverse set of classifiers, we experiment with the following classifiers for structured classification tasks: Logistic Classifier, MLP Classifier, and Decision Tree Classifier. For text-based tasks, we include BERT-base-uncased(Devlin et al., 2019) and RoBERTa-base classifiers(Liu et al., 2019).

To accurately evaluate the explanation generation capabilities of OPEX , we compare its explanations against those produced by several baseline methods. Across all tasks, we compare OPEX with explanations generated by `MaNtLE`, `LlaMA-3.1-8B Instruct`, `LlaMA-3.3-70B Instruct`, `GPT-4o`, and `DeepSeek-R1-Distill-Llama-70B` (zero-shot and few-shot). By comparing against various explanation generation methods, we evaluate the extent to which our training procedure enhances the explanation generation capabilities of OPEX . All experiments across both structured and text-based datasets use the same recoverability and simulatability evaluation protocol described in § 3.3.

## 5. Results and Analyses

**Explaining Model Predictions.** We first evaluate explanation quality when models (§ 4) are trained directly on the corresponding datasets, with results reported in Tables 2 and 8. Across all datasets, OPEX and its variants achieve near-perfect recoverability, significantly outperforming (paired t-test, $p < 0.01$) strong LLM baselines. While training improves the recoverability of all methods, non-OPEX explanations continue to exhibit substantially lower alignment

|  | Llama 8B | Llama 70B | GPT-4o | OPEX-R | OPEX-S | MaNtLE |
|---|---|---|---|---|---|---|
| ***Simulatability*** |  |  |  |  |  |  |
| Corona Sentiment | $0.46 \pm 0.12$ | $0.48 \pm 0.09$ | $0.47 \pm 0.10$ | $0.75 \pm 0.09$ | $\mathbf{0.76 \pm 0.09}$ | NA |
| YouTube Spam | $0.85 \pm 0.13$ | $0.94 \pm 0.05$ | $\mathbf{0.96 \pm 0.06}$ | $0.94 \pm 0.04$ | $0.95 \pm 0.04$ | $0.43 \pm 0.09$ |
| Iris | $0.82 \pm 0.11$ | $0.79 \pm 0.10$ | $0.94 \pm 0.03$ | $\mathbf{0.96 \pm 0.00}$ | $\mathbf{0.96 \pm 0.00}$ | $0.91 \pm 0.09$ |
| Car Evaluation | $0.68 \pm 0.11$ | $0.70 \pm 0.10$ | $0.75 \pm 0.09$ | $\mathbf{0.93 \pm 0.06}$ | $0.92 \pm 0.06$ | $0.66 \pm 0.16$ |
| Finance | $0.69 \pm 0.11$ | $0.75 \pm 0.11$ | $0.74 \pm 0.09$ | $\mathbf{1.00 \pm 0.00}$ | $\mathbf{1.00 \pm 0.00}$ | NA |
| Balance Scale | $0.62 \pm 0.16$ | $0.72 \pm 0.12$ | $0.61 \pm 0.15$ | $\mathbf{0.75 \pm 0.12}$ | $0.72 \pm 0.11$ | $0.35 \pm 0.25$ |
| *Avg* | $0.63 \pm 0.19$ | $0.68 \pm 0.19$ | $0.68 \pm 0.20$ | $\mathbf{0.86 \pm 0.13}$ | $\mathbf{0.86 \pm 0.13}$ | $0.52 \pm 0.22$ |
| ***Recoverability*** |  |  |  |  |  |  |
| Corona Sentiment | $0.50 \pm 0.12$ | $0.52 \pm 0.10$ | $0.51 \pm 0.10$ | $0.97 \pm 0.04$ | $\mathbf{0.98 \pm 0.03}$ | NA |
| YouTube Spam | $0.85 \pm 0.12$ | $0.95 \pm 0.05$ | $0.96 \pm 0.04$ | $\mathbf{1.00 \pm 0.01}$ | $\mathbf{1.00 \pm 0.00}$ | $0.44 \pm 0.11$ |
| Iris | $0.80 \pm 0.14$ | $0.81 \pm 0.04$ | $0.98 \pm 0.02$ | $\mathbf{1.00 \pm 0.00}$ | $\mathbf{1.00 \pm 0.00}$ | $0.90 \pm 0.13$ |
| Car Evaluation | $0.70 \pm 0.11$ | $0.81 \pm 0.08$ | $0.80 \pm 0.07$ | $\mathbf{0.98 \pm 0.02}$ | $\mathbf{0.98 \pm 0.03}$ | $0.67 \pm 0.11$ |
| Finance | $0.75 \pm 0.08$ | $0.81 \pm 0.10$ | $0.81 \pm 0.08$ | $\mathbf{1.00 \pm 0.00}$ | $\mathbf{1.00 \pm 0.00}$ | NA |
| Balance Scale | $0.66 \pm 0.14$ | $0.81 \pm 0.11$ | $0.85 \pm 0.07$ | $0.98 \pm 0.03$ | $\mathbf{0.99 \pm 0.02}$ | $0.38 \pm 0.23$ |
| *Avg* | $0.66 \pm 0.17$ | $0.73 \pm 0.18$ | $0.73 \pm 0.19$ | $\mathbf{0.99 \pm 0.03}$ | $\mathbf{0.99 \pm 0.03}$ | $0.53 \pm 0.20$ |

*Table 2.* Simulatability and Recoverability results for **explanations of classifier predictions on real datasets**. Best results per row are shown in bold. Scores are averaged over 40 runs and reported as mean $\pm$ standard deviation. We observe that OPEX explanations better simulate model decisions, as they capture the underlying classification rationale more effectively than the LLM baselines. Scores are averaged across Logistic, MLP, and Decision Tree classifiers for structured tasks, and BERT-base-uncased and RoBERTa-base classifiers for text-based tasks. MaNtLE explanations are only applicable to structured classification tasks.

with the learned decision boundaries of the underlying models. A similar pattern is observed for simulatability. OPEX explanations achieve significantly higher simulatability than all baselines (paired t-test, $p < 0.01$). Notably, OPEX maintains consistently high simulatability across datasets, even when baseline methods display increased variance. Overall, these results demonstrate that OPEX produces explanations that are not only faithful to the target model but also allow downstream users to accurately simulate model predictions across diverse datasets.

**Explaining Ground Truth Labels.** We next evaluate the performance of OPEX explanations against existing explanation methods on real-world datasets. As shown in Table 3, all methods are evaluated using recoverability and simulatability metrics. Across all datasets, OPEX and its variants again achieve near-perfect recoverability, significantly outperforming (paired t-test, $p < 0.01$) even the strongest LLM baselines. In contrast, all non-OPEX explanation frameworks exhibit substantially lower alignment with the underlying patterns present in the datasets. We observe a similar trend for simulatability. While larger LLMs improve over their smaller counterparts, OPEX explanations achieve significantly higher simulatability (paired t-test, $p < 0.01$) than the strongest baselines. Notably, OPEX models maintain high simulatability even when baseline methods exhibit substantial variance, indicating explanations that are both reliable and actionable on previously unseen samples.

**Synthetic Datasets.** This section provides a critical validation of our evaluation protocol, demonstrating that OPEX achieves high behavioral faithfulness even when the eval-

uator lacks relevant pretraining knowledge. We evaluate OPEX on a suite of synthetic tasks from (R. Menon et al., 2022) to assess whether its performance could be attributed to information leakage from the User-proxy LLM (Fang et al., 2025). As shown in Table 4, OPEX and its variants consistently outperform all baseline methods across both recoverability and simulatability. Performance on synthetic tasks is comparable to that observed on real-world datasets, indicating that OPEX explanations remain robust even in controlled settings. These synthetic tasks are absent from the User-proxy LLM's pretraining data, ruling out memorization or latent task familiarity. Thus, the observed gains arise from the quality of OPEX explanations rather than implicit knowledge in the User-proxy LLM. Overall, the results show that OPEX produces genuinely faithful explanations, independent of information leakage.

**Comparison with Human-Written Explanations.** We compare OPEX explanations with human-written explanations using simulatability on the Car Evaluation dataset (R. Menon et al., 2022). Human-written explanations achieve $0.45 \pm 0.10$ simulatability, while OPEX-R reaches $0.92 \pm 0.05$. This two-fold gain indicates that human explanations often omit details necessary for accurate simulation, whereas OPEX explanations are optimized to capture the model's decision logic. We evaluate only on Car Evaluation because all other datasets with human-written explanations were used during OPEX 's training.

**Impact of Model Size.** We analyze the impact of model size on explanation quality by varying the number of parameters in OPEX . To this end, we introduce OPEX-SMALL ,

|  | Llama 8B | Llama 70B | GPT-4o | OPEX-R | OPEX-S |
|---|---|---|---|---|---|
| ***Simulatability*** | | | | | |
| Corona Sentiment | $0.38 \pm 0.10$ | $0.41 \pm 0.10$ | $0.41 \pm 0.11$ | $0.67 \pm 0.09$ | $\mathbf{0.68 \pm 0.09}$ |
| YouTube Spam | $0.80 \pm 0.14$ | $0.92 \pm 0.07$ | $\mathbf{0.95 \pm 0.04}$ | $0.94 \pm 0.05$ | $0.94 \pm 0.05$ |
| Iris | $0.73 \pm 0.15$ | $0.82 \pm 0.04$ | $\mathbf{0.93 \pm 0.05}$ | $0.92 \pm 0.04$ | $\mathbf{0.93 \pm 0.03}$ |
| Car Evaluation | $0.65 \pm 0.09$ | $0.69 \pm 0.10$ | $0.73 \pm 0.10$ | $\mathbf{0.92 \pm 0.05}$ | $\mathbf{0.92 \pm 0.05}$ |
| Finance | $0.66 \pm 0.12$ | $0.71 \pm 0.10$ | $0.71 \pm 0.10$ | $\mathbf{1.00 \pm 0.00}$ | $\mathbf{1.00 \pm 0.00}$ |
| Balance Scale | $0.57 \pm 0.18$ | $0.70 \pm 0.14$ | $0.66 \pm 0.13$ | $\mathbf{0.73 \pm 0.09}$ | $0.71 \pm 0.11$ |
| *Avg* | $0.58 \pm 0.20$ | $0.64 \pm 0.21$ | $0.65 \pm 0.22$ | $\mathbf{0.83 \pm 0.15}$ | $\mathbf{0.83 \pm 0.15}$ |
| ***Recoverability*** | | | | | |
| Corona Sentiment | $0.44 \pm 0.10$ | $0.43 \pm 0.10$ | $0.45 \pm 0.10$ | $\mathbf{0.98 \pm 0.03}$ | $\mathbf{0.98 \pm 0.03}$ |
| YouTube Spam | $0.85 \pm 0.13$ | $0.95 \pm 0.05$ | $0.96 \pm 0.04$ | $\mathbf{1.00 \pm 0.00}$ | $\mathbf{1.00 \pm 0.00}$ |
| Iris | $0.70 \pm 0.15$ | $0.83 \pm 0.06$ | $0.98 \pm 0.02$ | $\mathbf{1.00 \pm 0.00}$ | $\mathbf{1.00 \pm 0.00}$ |
| Car Evaluation | $0.69 \pm 0.11$ | $0.80 \pm 0.12$ | $0.80 \pm 0.09$ | $\mathbf{0.99 \pm 0.02}$ | $\mathbf{0.99 \pm 0.02}$ |
| Finance | $0.74 \pm 0.11$ | $0.76 \pm 0.09$ | $0.78 \pm 0.10$ | $\mathbf{1.00 \pm 0.00}$ | $\mathbf{1.00 \pm 0.00}$ |
| Balance Scale | $0.62 \pm 0.14$ | $0.77 \pm 0.08$ | $0.83 \pm 0.08$ | $0.98 \pm 0.03$ | $\mathbf{0.99 \pm 0.02}$ |
| *Avg* | $0.63 \pm 0.19$ | $0.69 \pm 0.22$ | $0.71 \pm 0.22$ | $\mathbf{0.99 \pm 0.03}$ | $\mathbf{0.99 \pm 0.02}$ |

*Table 3.* Simulatability and Recoverability results on **explanations of real datasets with ground truth labels**. Best results per row are shown in bold. Scores are averaged over 40 runs and reported as mean $\pm$ standard deviation. LLM baselines typically generate fluent but underspecified explanations. While plausible to humans, such explanations usually omit decision criteria, resulting in low simulatability and recoverablity despite high linguistic quality.

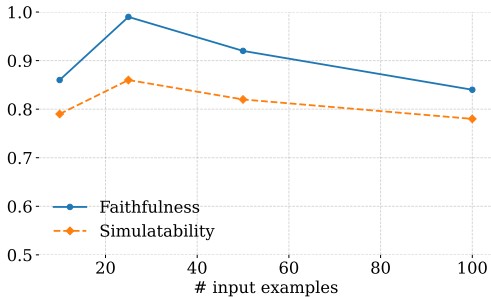

*Figure 2.* Effect of the number of input examples on explanation quality. Simulatability and Recoverability as a function of the number of labeled examples provided to OPEX . Both metrics peak at 25 examples, after which performance degrades, suggesting that additional inputs introduce noise rather than improving induction.

which is trained using the same GRPO procedure as OPEX-R , with the only change being the base model, replaced by `Llama-3.2-3B-Instruct`. This setup isolates the effect of model capacity by reducing the parameter count to 3B while keeping the training pipeline and objectives fixed.

Compared to OPEX-R and OPEX-S (Table 3), OPEX-SMALL achieves comparable recoverability across all datasets, with average recoverability remaining near-saturated (Table 5). This indicates that our training procedure enables OPEX-SMALL to align with the classification rationale of the target model, and that this alignment is preserved even under parameter reduction. We observe only a marginal drop in simulatability, from $0.86$ to $0.85$, relative to larger variants. Despite this, OPEX-SMALL still substantially outperforms all baseline explanation methods.

Overall, the results show that our training procedure enables a compact 3B-parameter model to produce faithful, simulatable, and robust explanations while generalizing to unseen tasks and samples with minimal performance degradation.

**Effect of Number of Samples.** We also vary the number of data samples provided as input to OPEX to understand the effect of the number of samples on the quality of explanations. As shown in Figure 2, we observe that explanation recoverability peaks at 25 samples (0.99) and declines as more samples are added. A similar trend is observed for simulatability, which also achieves its highest value at 25 samples (0.86). While increasing the sample size from 10 to 25 improves both metrics, providing more than 25 samples leads to diminishing returns and degraded performance. In particular, explanations generated with 100 samples exhibit lower recoverability and simulatability, suggesting that excessive inputs may introduce noise and hinder OPEX 's ability to capture the classifier's rationale and produce concise, effective explanations.

**Influence of the User-proxy LLM.** To ensure that the reported metrics are not artifacts of a particular evaluator model or a consequence of architectural similarities between OPEX (Llama-based) and the User-proxy LLM, we conduct an additional evaluation of OPEX explanations using `GPT-4o` as the User-proxy LLM. This cross-model evaluation allows us to assess whether the explanations genuinely encode the underlying classification rationale in a broadly comprehensible manner, rather than merely capitalizing on idiosyncratic features shared within the Llama model family.

The corresponding results are presented in Table 6. When

|  | Llama 8B | Llama 70B | GPT-4o | OPEX-R | OPEX-S |
|---|---|---|---|---|---|
| *Simulatability* | | | | | |
| Task 1 | 0.58 ± 0.07 | 0.79 ± 0.14 | 0.73 ± 0.09 | **0.90 ± 0.08** | 0.89 ± 0.09 |
| Task 2 | 0.54 ± 0.12 | 0.68 ± 0.14 | 0.68 ± 0.15 | **0.88 ± 0.09** | 0.87 ± 0.12 |
| Task 3 | 0.65 ± 0.11 | 0.87 ± 0.12 | 0.82 ± 0.14 | 0.90 ± 0.07 | **0.91 ± 0.07** |
| Task 4 | 0.56 ± 0.09 | 0.76 ± 0.12 | 0.70 ± 0.12 | **0.82 ± 0.09** | 0.80 ± 0.10 |
| *Avg* | 0.58 ± 0.11 | 0.78 ± 0.15 | 0.73 ± 0.14 | **0.88 ± 0.09** | 0.87 ± 0.10 |
| *Recoverability* | | | | | |
| Task 1 | 0.65 ± 0.07 | 0.87 ± 0.12 | 0.81 ± 0.09 | **1.00 ± 0.01** | **1.00 ± 0.01** |
| Task 2 | 0.64 ± 0.08 | 0.81 ± 0.11 | 0.80 ± 0.10 | **0.99 ± 0.02** | 0.99 ± 0.03 |
| Task 3 | 0.74 ± 0.05 | 0.94 ± 0.06 | 0.93 ± 0.06 | **0.99 ± 0.01** | 0.99 ± 0.02 |
| Task 4 | 0.69 ± 0.06 | 0.86 ± 0.08 | 0.87 ± 0.07 | 0.95 ± 0.04 | **0.96 ± 0.03** |
| *Avg* | 0.68 ± 0.08 | 0.87 ± 0.11 | 0.85 ± 0.09 | **0.98 ± 0.03** | **0.98 ± 0.03** |

*Table 4.* Simulatability and Recoverability results of **synthetic tasks** from R. Menon et al. (2022). Scores are averaged over 40 runs and reported as mean ± standard deviation. The strong performance of OPEX explanations indicates that the observed results are not due to knowledge leakage from the user-proxy LLM.

|  | *Simulatability* | *Recoverability* |
|---|---|---|
| Corona Sentiment | 0.74 ± 0.08 | 0.98 ± 0.03 |
| YouTube Spam | 0.91 ± 0.08 | 1.00 ± 0.00 |
| Iris | 0.96 ± 0.03 | 1.00 ± 0.00 |
| Car Evaluation | 0.93 ± 0.05 | 0.99 ± 0.02 |
| Finance | 1.00 ± 0.02 | 0.99 ± 0.06 |
| Balance Scale | 0.72 ± 0.09 | 0.98 ± 0.03 |
| Avg | 0.85 ± 0.13 | 0.99 ± 0.03 |

*Table 5.* Impact of model size reduction on explanation quality using OPEX-SMALL (3B parameters). Both metrics degrade only modestly compared to OPEX (8B parameters), indicating that large explanation models are not required for behavioral faithfulness. Scores are averaged over 40 runs and reported as mean ± stdev.

|  | OPEX-R | | OPEX-S | |
|---|---|---|---|---|
| Dataset | Recov. | Sim. | Recov. | Sim. |
| Corona Sentiment | 0.97 | 0.68 | 0.97 | 0.67 |
| YouTube Spam | 1.00 | 0.95 | 1.00 | 0.96 |
| Iris | 1.00 | 0.93 | 0.98 | 0.94 |
| Car Evaluation | 1.00 | 0.92 | 1.00 | 0.91 |
| Finance | 1.00 | 1.00 | 1.00 | 1.00 |
| Balance Scale | 0.98 | 0.72 | 0.98 | 0.71 |
| Avg | 0.99 | 0.84 | 0.99 | 0.84 |

*Table 6.* Evaluator robustness of OPEX explanations. Recoverability and simulatability of explanations from OPEX-R and OPEX-S when evaluated using `GPT-4o` as the User-proxy LLM. Results closely match evaluation with `LLaMA-3.3-70B-Instruct`, suggesting that explanations capture transferable model behavior rather than evaluator-specific artifacts.

evaluated with `GPT-4o`, OPEX-R and OPEX-S attain an average recoverability of 0.99 and an average simulatability of 0.84. These outcomes are closely aligned with our primary evaluation using `Llama-3.3-70B-Instruct`: the average recoverability remains unchanged, suggesting that OPEX explanations are consistently interpreted as faithful reflections of the classifier's behavior irrespective of the choice of evaluator. We observe only a minor decrease in simulatability, from 0.86 (Table 2) to 0.84 (Table 6). This stability across distinct user-proxy LLMs suggests that OPEX generates robust transferable explanations that reliably capture classifiers' decision-making logic.

**Comparing with CoT Based Explanations.** OPEX explanations substantially outperform Chain-of-Thought (CoT) baselines in behavioral faithfulness across real-world datasets. While CoT explanations generated by `DeepSeek-R1-Distill-Llama-70B` (with and without few-shot prompting) achieve moderate recoverability (0.73–0.76) and simulatability (0.65–0.66), they exhibit high variance and inconsistent generalization, particularly on text-based tasks. In contrast, OPEX achieves near-

perfect recoverability and consistently higher simulatability, with improvements that are statistically significant (paired t-test, $p < 0.01$). These results suggest that prompting-based reasoning alone is insufficient for faithful explanations, whereas directly optimizing for behavioral objectives yields explanations that more reliably capture and generalize model decision logic.

**User Study.** To assess the practical utility of OPEX explanations, we conducted a user study measuring how effectively they help users classify unseen samples. We quantify improvements in human accuracy when participants are provided with explanations from OPEX-R compared to alternative explanation models, and analyze qualitative differences in explanation quality.

We recruited 60 participants via Prolific and compensated them at 12 USD per hour. The study used the Car Evaluation dataset and followed a between-subjects design. Participants were randomly assigned to one of

| Dataset | CoT Recov. | CoT Sim. | CoT + Few-shot Recov. | CoT + Few-shot Sim. | OPEX Recov. | OPEX Sim. |
|---|---|---|---|---|---|---|
| Corona Sentiment | 0.52 | 0.45 | 0.51 | 0.44 | **0.98** | **0.68** |
| YouTube Spam | 0.96 | **0.95** | **1.00** | 0.94 | **1.00** | **0.95** |
| Iris | 0.99 | 0.82 | **1.00** | 0.82 | **1.00** | **0.93** |
| Car Evaluation | 0.77 | 0.71 | 0.82 | 0.73 | **0.99** | **0.92** |
| Finance | 0.87 | 0.72 | 0.92 | 0.74 | **1.00** | **1.00** |
| Balance Scale | 0.75 | 0.70 | 0.91 | 0.65 | **0.98** | **0.72** |
| Avg | 0.73 | 0.66 | 0.76 | 0.65 | **0.99** | **0.84** |

*Table 7.* Comparison of OPEX with Chain-of-Thought (CoT) explanations generated using `DeepSeek-R1-Distill-Llama-70B`. OPEX significantly outperforms both CoT and CoT + few-shot baselines in recoverability and simulatability across datasets (paired $t$-test, $p < 0.01$), demonstrating the advantage of directly optimizing explanations for behavioral faithfulness.

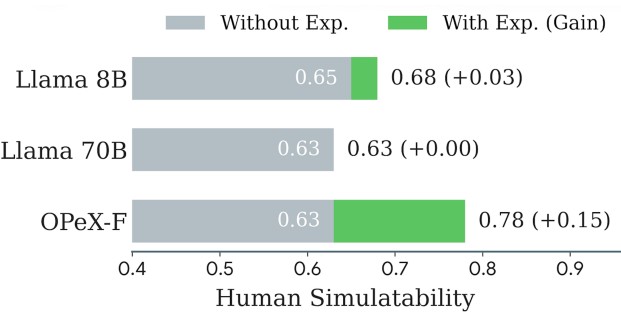

*Figure 3.* Human user study: effect of explanations on human simulatablity (classification accuracy on unseen samples) with and without explanations. OPEX-R explanations improve accuracy by 15 percentage points over no explanations. In contrast, providing explanations barely helps users for `LLaMA-3.1-8B-Instruct` and `LLaMA-3.3-70B-Instruct`.

three groups (20 per group), corresponding to explanations from OPEX-R, `LLaMA-3.3-70B-Instruct`, and `LLaMA-3.1-8B-Instruct`. Each session comprised two phases: participants first classified 20 samples without explanations to establish a baseline, and then classified the same 20 samples with explanations from their assigned model. We measured accuracy in both phases and computed the performance improvement.

As shown in Figure 3, OPEX-R yields the largest accuracy gain, improving performance from 0.63 to 0.78 (+0.15). In comparison, `LLaMA-3.1-8B-Instruct` explanations produce only a modest improvement, from 0.65 to 0.68 (+0.03), while `LLaMA-3.3-70B-Instruct` explanations provide no measurable benefit, with accuracy remaining at 0.63. These results show that OPEX-R substantially outperforms general-purpose LLM explanations, including larger models, in enabling users to simulate classifier behavior on unseen data. Participant feedback further reveals qualitative differences in explanation utility. Users reported that `LLaMA-3.3-70B-Instruct` explanations were of-

ten verbose and focused on describing individual samples rather than conveying generalizable rules, increasing cognitive load without aiding transfer to new instances. In contrast, OPEX-R explanations were described as concise and actionable, clearly capturing the underlying classification logic. This feedback aligns with the observed accuracy gains from OPEX explanations. Additional details about the user study can be found in § A.4

## 6. Conclusion

This work reframes natural-language explanations as functional artifacts whose value lies in their ability to reproduce and predict model behavior. We formalize this perspective through *behavioral faithfulness*, operationalized via recoverability and simulatability, two metrics that test whether an explanation can act as a surrogate for a classifier's decision function. By optimizing explanations for these objectives rather than human-written rationales, OPEX removes a key bottleneck in explainability and produces concise, testable, and generalizable explanations.

Empirically, OPEX consistently outperforms substantially larger models and improves human users' ability to anticipate model predictions. These gains also hold against rationale-based explanation systems such as MaNtLE and Chain-of-Thought based approaches, indicating that directly optimizing for behavioral faithfulness yields explanations that are more predictive and generalizable than plausibility-oriented rationales. More broadly, we frame explanation generation as an optimization problem with explicit, testable objectives, opening the door to richer explanation learning. This includes extending behavioral faithfulness to sequential decision-making, uncertainty or counterfactual objectives, and hybrid behavioral–mechanistic signals—shifting the field from plausibility-driven explanations toward explanations defined by what they enable.

## 7. Limitations

In this work, we restrict OPEX explanations to structured and text-based classification tasks. Generation and multi-modal based tasks remain outside the scope of this work. Thus, it is not yet clear how well our behavioral-faithfulness objectives transfer to open-ended generation or settings where explanations must account for interactions among multiple models. However, we believe the same framework would hold, as RL rewards tend to be effective even when the task is open-ended. In addition, our evaluation relies on a user-proxy LLM for scalable reward computation and automatic assessment, which may not capture all aspects of human judgment. Finally, the human study is conducted on a single dataset, so broader user evaluation across more domains remains an important direction for future work.

## Impact Statement

This paper presents work aimed at advancing the field of interpretable machine learning by improving the recoverability and usability of natural language explanations. By enabling explanations that more accurately reflect model behavior, this work has the potential to support transparency, accountability, and informed decision-making in high-stakes applications. We do not foresee significant negative societal impacts beyond those already associated with the deployment of machine learning systems.

## Acknowledgements

The authors would like to thank the anonymous reviewers for their suggestions and feedback on the work. This work was supported in part by a gift from CoefficientGiving.

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

## A. Appendix

### A.1. Training and Optimization Details

All models were trained using NVIDIA A6000 GPUs with CUDA enabled. We fine-tuned a `Llama-3.1-8B-Instruct` base model using parameter-efficient LoRA adaptation. Training details are summarized below.

- **Base model:** `Llama-3.1-8B-Instruct`

- **Hardware:** NVIDIA A6000 GPUs

- **Sequence length:** Maximum sequence length of 8,000 tokens

- **Prompt length:** Maximum prompt length of 4,000 tokens

- **LoRA configuration:** Rank 16 with scaling factor 16

- **LoRA target modules:** `q_proj`, `k_proj`, `v_proj`, `o_proj`, `gate_proj`, `up_proj`, `down_proj`

- **LoRA layers:** Applied to all transformer layers

- **Gradient checkpointing:** Enabled via Unsloth for long-context training

- **Optimizer:** Paged AdamW (8-bit)

- **Learning rate:** $5 \times 10^{-5}$

- **Learning rate schedule:** Cosine decay with warmup ratio 0.1

- **Adam parameters:** $\beta_1 = 0.9$, $\beta_2 = 0.99$

- **Weight decay:** 0.1

- **Batch size:** 2 per device

- **Gradient accumulation:** 1 step

- **Gradient clipping:** Maximum norm of 0.1

- **Training epochs:** 1

- **Logging:** Weights & Biases

- **Checkpointing:** LoRA weights merged and saved as 16-bit model checkpoints

| | Llama 8B | Llama 70B | GPT 4o | OPEX-R | OPEX-S |
|---|---|---|---|---|---|
| **Simulatability** | | | | | |
| Decision Tree | 0.56 | 0.64 | **0.68** | **0.68** | **0.68** |
| Logistic | 0.84 | 0.72 | 0.68 | **1.00** | **1.00** |
| MLP | 0.88 | 0.80 | 0.92 | **0.96** | **0.96** |
| BERT | 0.68 | 0.56 | 0.68 | **0.76** | **0.76** |
| RoBERTa | **0.64** | 0.60 | 0.56 | 0.56 | 0.56 |
| *Avg* | 0.62 | 0.67 | 0.67 | **0.86** | **0.86** |
| **Recoverability** | | | | | |
| Decision Tree | 0.52 | 0.92 | 0.80 | **1.00** | **1.00** |
| Logistic | 0.64 | 0.80 | 0.76 | **1.00** | **1.00** |
| MLP | 0.88 | 0.88 | **1.00** | **1.00** | **1.00** |
| BERT | 0.44 | 0.52 | 0.68 | 0.96 | **1.00** |
| RoBERTa | 0.52 | 0.64 | 0.68 | 0.96 | **1.00** |
| *Avg* | 0.66 | 0.73 | 0.73 | 0.98 | **1.00** |

*Table 8.* Simulatability and Recoverability results grouped by downstream classifier. Best results per row are shown in bold. Results are averaged over 40 runs across all tasks. We observe that OPEX explanations are more faithful to the classifiers' decision processes than those produced by other approaches.

## A.2. Prompts and Additional Experiments

## A.3. Examples of OPEX Explanation

**Corona Sentiment.** The 'target' variable is determined by a complex interplay of various input features, with a strong emphasis on the tone and sentiment expressed in the text. When the text contains words or phrases with negative connotations, such as "bad hygiene," "arrogant tosser," or "fake news media," the 'target' variable tends to be extremely negative, indicating a strong sense of criticism or disapproval. On the other hand, when the text expresses gratitude, appreciation, or positivity, such as "thanks," "leadership," or "blessings," the 'target' variable is extremely positive, indicating a strong sense of admiration or approval. The presence of words related to the COVID-19 pandemic, such as "coronavirus," "COVID-19," or "pandemic," tends to have a neutral or slightly negative effect on the 'target' variable, but this can be mitigated by the presence of words with positive connotations, such as "helping those in need" or "essential products." The 'target' variable is also influenced by the tone of the text, with a more formal or professional tone tending to result in a more positive 'target' variable, while a more casual or informal tone tends to result in a more negative 'target' variable. Additionally, the presence of words related to shopping, such as "grocery store," "supermarket," or "online delivery," tends to have a positive effect on the 'target' variable, especially when combined with words related to convenience or accessibility, such as "click and collect" or "in-store pickup." Overall, the 'target' variable is determined by a complex interplay of tone, sentiment, and context, with a strong emphasis on the expression of gratitude, appreciation, or positivity.

---

**Prompt Template for Faithful Explanation Generation**

Given the following data samples:
`{data_samples}`

Generate a single, comprehensive natural-language paragraph that explains how the input features determine the `target` variable. The explanation must be based only on the provided data samples.

**Important:**

- The explanation must describe how all combinations of features, including their values and ranges, determine the `target` variable, and it must be sufficient to predict the `target` for new, unseen samples.

- Ensure the explanation is exhaustive, captures all underlying patterns in the data, and generalizes beyond the given samples.

- Do not refer to any model, predictions, or training process, and do not use real-world or external knowledge.

- Use absolute values and explicit ranges; avoid relative or qualitative terms.

- Output only a single, complete paragraph. Do not include code, warnings, headings, or any additional text.

*Table 9.* Prompt used to generate faithful, generalizable natural-language explanations from data samples.

**YouTube Spam.** The 'target' variable is determined by a combination of the AUTHOR's behavior, the DATE of the comment, and the CONTENT of the comment. When the AUTHOR is asking for a subscription or promotion, such as by saying "subscribe my channel" or "check out this video," and the DATE is within a certain range, typically between 2013 and 2015, the 'target' is likely to be 1. This is because these comments are often spam or self-promotional, and the AUTHOR is trying to increase their visibility or engagement. On the other hand, when the AUTHOR is commenting on the content of the video, such as by saying "this song is nice" or "why so many downs," and the DATE is also within the same range, the 'target' is likely to be 0. This is because these comments are often genuine and focused on the content of the video. However, when the AUTHOR is promoting a product or service, such as by saying "make money online" or "visit this website," the 'target' is also likely to be 1, regardless of the DATE. This is because these

comments are often spam or promotional, and the AUTHOR is trying to sell something. Overall, the 'target' variable is determined by a combination of the AUTHOR's behavior, the DATE, and the CONTENT, and can be predicted based on these factors.

**Iris.** The 'target' variable is primarily determined by the petal length and petal width, with sepal length and sepal width playing a secondary role. Petal length values above 4.5 tend to correspond to Iris-virginica, while values below 1.4 are more likely to be Iris-setosa. Petal width values above 1.9 are also characteristic of Iris-virginica, whereas values below 0.2 are typical of Iris-setosa. Sepal length values above 6.0 and sepal width values above 3.2 are more commonly associated with Iris-virginica, whereas values below 5.0 and 3.0, respectively, are more indicative of Iris-setosa. The combination of petal length and petal width is the most decisive factor, with petal length and petal width values above 4.5 and 1.9, respectively, strongly suggesting Iris-virginica.

**Car Evaluation.** The 'target' variable is primarily determined by the combination of 'buying' and 'safety' features, with 'buying' having a significant impact when its value is 'low' or 'high', and 'safety' having a significant impact when its value is 'high'. The 'doors' feature also plays a crucial role, with a higher number of doors generally corresponding to a lower 'target' value, particularly when 'doors' is 4 or more. The 'persons' feature is also influential, with a higher number of persons generally corresponding to a lower 'target' value, especially when 'persons' is 4 or more. The 'lug_boot' feature has a moderate impact, with a 'big' lug boot generally corresponding to a lower 'target' value, and a 'med' lug boot generally corresponding to a higher 'target' value. The 'maint' feature has a moderate impact, with a 'high' maintenance cost generally corresponding to a lower 'target' value, and a 'low' maintenance cost generally corresponding to a higher 'target' value.

**Finance.** The 'target' variable is determined by a combination of factors, including the direction and magnitude of changes in various metrics such as stock prices, revenue, and profit, as well as the presence or absence of specific events or transactions. A 'not neutral' outcome is more likely when there are significant changes in stock prices, such as a 0.87% decrease, or when there are notable increases in revenue, such as a 5% or 4% increase. Additionally, a 'not neutral' outcome is also associated with the presence of specific events or transactions, such as the signing of a contract, the disposal of assets, or the reshuffling of executives. In contrast, a 'neutral' outcome is more likely when there are no significant changes in stock prices or revenue, or when there are no notable events or transactions. The magnitude of the change in stock prices or revenue also plays a role, with

larger changes being more likely to result in a 'not neutral' outcome. Furthermore, the type of event or transaction also matters, with certain types, such as the signing of a contract or the disposal of assets, being more likely to result in a 'not neutral' outcome.

**Balance Scale.** The 'target' variable is determined by a complex interplay of the input features, with the right-distance and left-distance features playing a crucial role in determining the outcome. When the right-distance is greater than 3, the target is more likely to be R, while a left-distance greater than 4 tends to result in a target of L. However, when the right-distance is less than 2, the target is more likely to be L, regardless of the left-distance. The right-weight and left-weight features also have a significant impact, with higher values above 4 often resulting in a target of R, while lower values below 2 tend to result in a target of L. Notably, when the right-weight is 5 and the left-weight is 4, the target is often R, but when the left-weight is 5, the target is often L. Additionally, when the right-distance is 3 and the left-distance is 4, the target is often L, but when the left-distance is 3, the target is often R. Overall, the 'target' variable is influenced by a combination of the values and ranges of the input features, with no single feature or combination of features being decisive in all cases.

### A.4. Additional Details on User Study

The user study was administered using an online survey. Participants were recruited through Prolific and were required to confirm that they were comfortable reading and understanding written English and had at least a high school degree. The participants were compensated at 12 USD per hour. Prior to participation, all users provided informed consent and were informed that their responses would be anonymized and used solely for research purposes.

The study consisted of two main phases. In the first phase, participants classified 20 samples from the Car Evaluation dataset using only the provided feature attributes. Each sample contained six categorical attributes describing a car: *buying*, *maint*, *doors*, *persons*, *lug_boot*, and *safety*. Participants selected one of four class labels: *unacc*, *acc*, *good*, or *vgood*. This phase established a baseline measure of participant performance without explanations.

In the second phase, participants classified the same 20 samples again, but this time accompanied by explanations generated by the explanation model assigned to their experimental group. Participants were explicitly instructed to classify the samples according to the explanations provided, even if the explanations appeared counterintuitive or inconsistent with real-world reasoning. To account for explanations that participants could not interpret, we additionally provided an "Unable to Decide" option.

**Classification Task Instructions**

In this section, you will classify examples from a car evaluation task **without seeing any explanation**.

- Each example consists of a set of attributes describing a car.

- Your task is to classify each car **based only on the attributes shown**.

- Please read each example carefully and select the option that best represents the **overall acceptability** of the car.

- There are no trick questions. We are interested in your **judgment and understanding, not speed**.

*Table 10.* Classification Task Instructions

The study employed a between-subjects design with three explanation conditions corresponding to explanations generated by OPEX , LLaMA-3.3-70B-Instruct, and LLaMA-3.1-8B-Instruct. Participants were randomly assigned to one condition and only viewed explanations from their assigned model throughout the study. Each participant therefore completed 40 classification decisions in total: 20 without explanations and 20 with explanations. To reduce confounding factors, the order and wording of the classification tasks were kept consistent across conditions, with the only difference being the explanation model shown during the second phase. Participants were also informed that the study focused on understanding how explanations assist classification rather than evaluating prior domain knowledge about cars. At the conclusion of the study, participants were given an optional free-response feedback section where they could provide comments regarding the quality of the explanation and whether the explanation enabled them to better classify the samples.

---

**Car Evaluation Task & Response Instructions**

---

**Car Evaluation Task**

Each example describes a car using the following attributes:

- **buying**: purchase price

- **maint**: maintenance cost

- **doors**: number of doors

- **persons**: passenger capacity

- **lug_boot**: luggage boot size

- **safety**: safety level

Based on these attributes, your task is to determine the **overall acceptability** of the car.

The possible class labels are:

- **unacc** — unacceptable

- **acc** — acceptable

- **good** — good

- **vgood** — very good

**How to Respond**

- **Select exactly one option** from the four labels above.

- Choose the option that best matches your judgment based on the information shown.

- Please answer all questions honestly and to the best of your ability.

---

*Table 11.* Car Evaluation Task and Response Instructions

---

**Explanation Guided Classification Task Instructions**

---

In this section, you will see the same car examples again, this time together with an explanation.

- For each example, your task is to classify the car **after reading the explanation provided**.

- Use the explanation to inform your decision.

- When classifying samples using the explanation, **do not rely on real-world knowledge**.

- Please classify the samples as closely as possible in accordance with the explanation.

**NOTE:**
Please classify the sample as described in the explanations, even if the explanations seem wrong, illogical, or counterintuitive.
If you are unable to classify the sample using the explanation, please select: **"Unable to Decide"**

- Please read both the attributes and the explanation carefully before responding.

- As before, there are no trick questions. We are interested in your **judgment and understanding, not speed**.

---

*Table 12.* Explanation Guided Classification Instructions

---

**Optional Feedback**

---

Please provide any feedback regarding the explanations shown during the study.

In particular, we are interested in understanding:

- Whether the explanations helped you classify the samples more accurately or confidently.

- Whether the explanations were clear, concise, and easy to follow.

- Any aspects of the explanations that you found confusing, misleading, repetitive, or unhelpful.

- Any additional comments about the overall study experience.

**Your feedback is optional and will be used only for research purposes.**

---

*Table 13.* Optional Feedback Prompt

