# OpenReview forum: "Directly Optimizing Natural Language Explanations for Behavioral Faithfulness: Simulatability and Recoverability"
_ICML.cc/2026/Conference — ICML 2026 regular_

### Official Review · Reviewer_buhF · 2026-03-10

**Soundness:** 2
**Presentation:** 2
**Significance:** 3
**Originality:** 2
**Overall Recommendation:** 5
**Confidence:** 4

**Summary:**

This paper trains surrogate explanation models that generate explanations without directly imitating gold explanations. Instead of supervised learning on reference explanations—which can encourage plausibly sounding but unfaithful rationalizations—the model is trained using rewards based on behavioral faithfulness, specifically recoverability and simulatability. The results suggest that this training approach yields more faithful explanations, and that it remains effective even when using relatively small models ( at 8B parameters).

**Compliance With Llm Reviewing Policy:**

Affirmed.

**Final Justification:**

TLDR: Updated my score from 2 to 5. Really nice to see a discussion round do its job. :)

I am honestly impressed with this rebuttal. The rigorous description of the metrics was much-needed, and assuming the authors incorporate this and the promised clarifications (re: other weaknesses) into the final version, this is a clear accept from me.

Full transparency, I was considering after reading Reviewer BPNE's review to raise to 4 (and not to 5), but you resolved their W1 weakness with extra experiments, really nice. Regarding the task scope (Reviewer BPNE W3): Going beyond classification would have made the paper "cleaner" but I don’t see the current scope as a dealbreaker. From my own experience working with RL rewards, they tend to be effective whether the task is open-ended or not. I'm satisfied with the results as presented.

**Key Questions For Authors:**

How are simulatability and recoverability exactly implemented?

**Limitations:**

yes

**Strengths And Weaknesses:**

**Post-rebuttal: I've raised from 2 to 5, see "rebuttal acknowledgement" for reasons.**

# Strengths

* **Interesting training objective.** The paper proposes generating explanations by optimizing for behavioral faithfulness (recoverability and simulatability) rather than imitating human-written explanations. This addresses a known issue with supervised explanation training, which can encourage plausible-sounding but unfaithful rationalizations. Yet, it is quite unclear to me how "simulatability" and "recoverability" are computed / defined / implemented. The principle though, sounds convincing. Also, the core idea—producing explanations that help (another) model recover the correct answer—resembles the motivation behind reasoning traces, though here it is framed as post-hoc explanation generation rather than intermediate reasoning. As such, the conceptual novelty may be somewhat limited.

* **Strong empirical results.** The reported results appear convincing and suggest that the proposed training approach improves explanation quality even with relatively small models (e.g., 8B) -- quality in terms of simulatability and recoverability (which is unsurprising, given that RL optimises for this) but human evaluations also support the extra quality. The paper evaluates across multiple datasets and includes a reasonable set of baselines, which strengthens the empirical claims.

# Weaknesses

* **Behavioral faithfulness may be difficult to interpret.** Metrics such as recoverability and simulatability can be satisfied even when the explanation does not reflect the true causal reasons behind the model’s prediction. For instance, the answer might already be easily inferable from the explanation itself, allowing another model to recover the prediction without the explanation actually capturing the original model’s reasoning. The paper should clarify how such cases are mitigated.

* **Claim about mechanistic faithfulness (L024) could be nuanced.** The statement that mechanistic faithfulness is often infeasible for black-box or proprietary models is somewhat strong. There exist approaches that require only limited model access (e.g., logits). For example, Parcalabescu et al. (2024), On Measuring Faithfulness…", analyze model internals using Shapley-based methods while requiring only logits, which many APIs expose.

* STRONG **Definitions of recoverability and simulatability lack precision** (L029). It is unclear how the "recovery" of the prediction happens and how an explanation can transfer to unseen examples. This also holds for the explanations in L158, 179. In the current description, it is unclear how to implement / measure the crux metrics of the paper.

* The notion of “explanation” is not clearly defined early enough. In the literature, reasoning traces (produced before the answer) are sometimes treated as explanations. Based on Figure 1, this paper appears to focus on post-hoc explanations, but this distinction should be clarified early in the paper to avoid confusion.

---

> ### Author Rebuttal · Authors · 2026-03-30
>
> We thank Reviewer buhF for the thoughtful feedback. We agree that the paper did not define its two core metrics precisely enough, and that this likely drove much of the confusion. We address this first, then the remaining concerns.
>
> **W1 (Strong): Definitions of recoverability and simulatability lack precision.**
> We will fix this by adding an explicit step-by-step algorithm in §3.3. Given a target classifier C, we sample inducing examples S = {(x_i, y_i)} with y_i = C(x_i) and generate one task-level explanation E from S. We then evaluate the explanation E through a User-proxy without access to the classifier C:
> - *Recoverability* = fraction of inducing examples whose labels the User-proxy predicts correctly given E and each x_i (no access to C).
> - *Simulatability* = fraction of labels predicted correctly on a disjoint held-out set Ŝ (also using only E).
>
> This makes clear that simulatability is measured on unseen examples, that both metrics use the same prediction protocol, and that simulatability prevents trivial solutions (e.g., memorizing labels) since the explanation must support accurate prediction on disjoint samples.
>
> **W2: Behavioral faithfulness may not reflect causal reasons.**
> We agree that behavioral faithfulness is a functional notion and does not by itself establish internal causal/mechanistic faithfulness. Our claim is intentionally narrower: OPeX is optimized to produce explanations that support accurate prediction of a classifier's observable behavior. We mitigate trivial shortcuts via (i) evaluation on unseen inputs (OPeX achieves 0.86 avg simulatability over 40 runs across tasks and modalities), (ii) synthetic tasks that reduce prior-knowledge leakage (Table 4), (iii) cross-evaluator validation with GPT-4o ruling out evaluator-specific shortcuts (Table 6), and (iv) a human study showing 15% accuracy improvement on unseen samples. We will revise the paper to state more explicitly that behavioral faithfulness is complementary to, not a replacement for, mechanistic faithfulness.
>
> **W3: Mechanistic faithfulness claim too strong.**
> We agree. Approaches like Parcalabescu et al. (2024) approximate mechanistic faithfulness with limited access. We will soften to "more difficult and less accessible to end users" and position behavioral faithfulness as complementary.
>
> **W4: "Explanation" not defined early enough; confusion with reasoning traces.**
> We will add in §1: "By explanation, we mean a post-hoc, task-level natural-language description of a classifier's decision behavior, generated after observing its input–output behavior. This is distinct from reasoning traces (e.g., CoT), which precede predictions during inference."
>
> **W5: Conceptual novelty limited—resembles reasoning traces.**
> Reasoning traces are (1) self-generated, (2) pre-prediction, (3) aimed at the model's own performance. OPeX explanations are (1) generated by a separate model about another model, (2) post-hoc, (3) optimized for third-party prediction. We can also support this empirically: in newly run baselines, DeepSeek-R1-Distill-Llama-70B with CoT reasoning achieves only 0.66 avg simulatability vs. OPeX's 0.86 (p < 0.01). This suggests that reasoning-trace-style approaches do not achieve the same behavioral faithfulness.
>
> **Key question: How are metrics exactly implemented?**
> Addressed in W1 above.
>
> We genuinely value your rigorous review, which has substantially enhanced the clarity and robustness of our paper! Please let us know if you have additional concerns. Given these clarifications, we also invite you to reconsider your evaluation scores.

---

> > ### Author Rebuttal · Reviewer_buhF · 2026-04-01
> >
> > TLDR: Updated my score from 2 to 5. Really nice to see a discussion round do its job. :)
> >
> > I am honestly impressed with this rebuttal. The rigorous description of the metrics was much-needed, and assuming the authors incorporate this and the promised clarifications (re: other weaknesses) into the final version, this is a clear accept from me.
> >
> > Full transparency, I was considering after reading Reviewer BPNE's review to raise to 4 (and not to 5), but you resolved their W1 weakness with extra experiments, really nice. Regarding the task scope (Reviewer BPNE W3): Going beyond classification would have made the paper "cleaner" but I don’t see the current scope as a dealbreaker. From my own experience working with RL rewards, they tend to be effective whether the task is open-ended or not. I'm satisfied with the results as presented.

---

### Official Review · Reviewer_NR5H · 2026-03-12

**Soundness:** 3
**Presentation:** 3
**Significance:** 3
**Originality:** 4
**Overall Recommendation:** 5
**Confidence:** 4

**Summary:**

This submission proposes to fine-tune LLMs using reinforcement learning to provide more behaviorally faithful natural language explanations of an ML classifier's predictions. Behavioral faithfulness is measured by the ability of another agent (human or LLM proxy) to predict the outputs of the classifier given only the explanation, on the same data that was used for fine-tuning (recoverability), or on unseen data (simulatability). Explanations are provided at the task level, i.e., they aim to explain the classifier's behavior across multiple examples of the task, rather than at the individual instance level. The proposed OPeX explainers are evaluated on tabular, text, and synthetic classification tasks (all not used in training) in comparison to simply prompting LLMs to give natural language explanations. A human user study is also performed on one of the classification tasks. The results show that OPeX achieves significantly higher recoverability and simulatability.

**Compliance With Llm Reviewing Policy:**

Affirmed.

**Final Justification:**

I was largely happy with the submission in my initial review. The rebuttal answered many of my questions and at least acknowledged my "Soundness" concerns 1 and 2 as limitations. The one aspect that I am less satisfied with is the positioning of explanations for ground truth labels. Overall, I maintain my assessment of Accept.

**Key Questions For Authors:**

1. Line 218: Why are recoverability and simulatability rewards scaled to $[-1, 1]$? Are they not already normalized to $[0, 1]$ since they are predictive accuracies?
1. In Tables 2 and 3, why does the "Avg" row (average over datasets) show larger standard deviations than rows for individual datasets? Shouldn't averaging reduce the standard deviation?
1. Why are two variants of OPeX needed, one optimized for recoverability (OPeX-F) and one for simulatability (OPeX-S)? It seems from Tables 2 and 3 that OPeX-F is almost as good at simulation as OPeX-S, and vice versa.
1. Since the degradation of the 3B-parameter OPeX model is minimal compared to the 8B variants, I wonder why the 3B model is not promoted at least as much as the 8B models? Indeed, Table 5 could have been just another column in Table 2 or 3.

**Limitations:**

I think the limitation to classification should be explicitly acknowledged.

**Strengths And Weaknesses:**

### Strengths
- I think the idea of directly optimizing the behavioral faithfulness of LLM-generated explanations is novel and valuable. I agree with the authors that natural language explanations should move beyond plausibility as an objective.
- The use of a user-proxy LLM allows recoverability and simulatability to be computed at scale for both RL training and evaluation.
- OPeX yields clear gains in recoverability and simulatability over LLM prompting baselines.
- I appreciate the additional experiments ensuring that the user-proxy LLM is not exerting an undue effect on the results, because of either leakage of its pre-trained knowledge or architectural similarity to the explaining LLM.
- The writing is generally very clear (if a bit repetitive)

### Weaknesses
Let me begin by saying that I do not have serious concerns. Below I categorize concerns by dimension. I also list questions that are not really weaknesses in the "Questions for Authors" section below.

Soundness:
1. In the human user study, why were the same 20 samples used in the second phase with explanations? Were participants told the outcomes from the first phase without explanations? Even if not, might participants have learned something about the samples from the first phase that would bias their results in the second phase?
1. None of the tabular datasets (at least those used for training) are high-dimensional in my view, with a maximum of 22 features. A related comment is that one of the evaluation datasets, Iris, is usually regarded as a toy dataset because it has only 150 samples and 4 features.
1. Why are explanations of ground truth labels evaluated in Table 2? Isn't the purpose of OPeX to explain classifier predictions, not ground truth? Why present this first in Section 5?

Presentation and contextualization:
1. Some elements in Figure 1 are unclear and never referenced in the text: $\theta_j$, $f(\theta_j)$, what is meant by "dataset trained on $f(\theta_j)$"
1. Under "Learning frameworks for explanation alignment" in Related Work, I think that the "amortization" of explanation methods, i.e., training a model to predict explanations given by a particular method, seems to be a rather relevant sub-area. Please see for example "Stochastic Amortization: A Unified Approach to Accelerate Feature and Data Attribution" by Covert et al. (NeurIPS 2024) and references therein. Amortization also allows generalization to unseen instances and could be viewed as a supervised alternative to the RL approach proposed in this work.
1. It is not clear how decision rules are described for text data. "Explicit feature conditions and value ranges" (line 137) may apply only to tabular data.
1. Some aspects about the classifiers are unclear:
    - Are multiple classifiers also used in training?
    - Which classifiers are used for the synthetic tasks?
    - Were the BERT and RoBERTa models not fine-tuned for classification?

Significance:
1. The approach is currently restricted to classification

Minor:
- It would help to mention in the introduction why "we employ a User-proxy LLM to compute reward signals," because recoverability and simulatability are traditionally evaluated by asking a human user to make predictions
- Line 201: The number of unseen test samples could be different from $N$, the number of training samples, right?
- The order of datasets in Tables 2, 3 etc. could be more principled, for example separating tabular and text datasets
- Lines 314-315: I think this sentence about allowing "downstream users to accurately simulate model predictions" should be deferred until after the human evaluation.
- Lines 362 and below: Should the reference be to Table 2 (ground truth labels) or 3 (model predictions)? If the former, then this does not agree with the phrase below about aligning with the "classification rationale of the target model."

---

> ### Author Rebuttal · Authors · 2026-03-30
>
> We thank Reviewer NR5H for the thorough and constructive review. We address each point below.
>
> **Q1: Human study—same 20 samples in both phases; possible learning bias?**
> We agree this is a limitation. We chose this design to keep task difficulty fixed in a within-subject comparison, and participants were not given feedback after Phase 1. Because the protocol was identical across all conditions, any familiarity effect should apply equally; however, only OPeX-F shows a substantial gain (+0.15, versus +0.03 and +0.00 for baselines). We will state this limitation explicitly and use disjoint or counterbalanced item sets in future versions.
>
> **Q2: Why evaluate ground truth labels (Table 2)?**
> Table 2 validates that OPeX can recover decision rules when they are noise-free, isolating explanation quality from classifier errors. It also enables comparison with human-written explanations. Our primary evaluation is Table 3 (classifier predictions). We will reorder to lead with Table 3 and reposition Table 2 as supporting analysis with clearer motivation.
>
> **S: Figure 1 elements unclear.**
> We agree. "Dataset trained on f(θ_j)" should read "dataset labeled by f(θ_j)." We will simplify notation and ensure all elements are defined and referenced.
>
> **S: Amortization literature.**
> We agree. Amortization (Covert et al., NeurIPS 2024) can be viewed as a supervised alternative: it trains a model to predict explanations from another method. OPeX differs in not relying on a predefined explanation target, instead optimizing directly for behavioral faithfulness. We will update Related Work.
>
> **S: Decision rules for text data.**
> For text tasks, the prompt captures semantic patterns (e.g., "reviews mentioning acting quality tend to be positive") rather than feature thresholds. We will clarify in §3.2 and include the text prompt in the appendix.
>
> **S: Dataset scale.**
> While these are not extreme high-dimensional structured tasks, they span tabular and text settings, multiple classifier families, and both real and synthetic domains. We will state this scope more carefully.
>
> **Q: Classifier details.**
> (a) Yes, multiple classifiers during training—each instance from one classifier/task. (b) Synthetic tasks use Logistic, MLP, and Decision Tree classifiers. (c) BERT and RoBERTa are fine-tuned for classification before use as targets. We will clarify.
>
> **Q: Why rewards scaled to [−1, 1]?**
> Scaling centers the distribution around zero for GRPO's group-normalization. With [0, 1] rewards, advantages are uniformly non-negative when explanations exceed random chance, reducing the contrastive signal between candidates. The [−1, 1] scaling ensures balanced positive/negative advantages, improving training stability.
>
> **Q: Why does "Avg" row show larger standard deviations?**
> The Avg row aggregates across datasets with different difficulty levels, introducing cross-dataset variability. Within-dataset deviations are small. We will add a clarifying note.
>
> **Q: Why two variants when performance is similar?**
> The similarity suggests recoverability is a strong proxy for simulatability in practice—an interesting finding. For deployment, either suffices. We will discuss this and consider recommending OPeX-F as default.
>
> **Q: Why not promote 3B model more?**
> We agree. We will integrate OPeX-SMALL into the main tables.
>
> **S: Minor presentation issues.**
> We will: (a) motivate User-proxy LLM in the introduction; (b) clarify test sample count can differ; (c) reorder tables by data type; (d) defer downstream-user claims until after human evaluation; (e) correct table reference to Table 3.
>
> We also note that in response to other reviewers' feedback, we ran additional baselines (MaNtLE, DeepSeek-R1-Distill-Llama-70B with CoT and CoT + few-shot). OPeX maintains significant advantages over all of them (p < 0.01), further supporting the contribution.
>
> We genuinely value your rigorous review, which has substantially enhanced the clarity and robustness of our paper! Please let us know if you have additional concerns.

---

> > ### Author Rebuttal · Reviewer_NR5H · 2026-04-03
> >
> > Thank you for the rebuttal and for answering my questions. My points 1 and 2 under "Soundness" were at least acknowledged as limitations.
> >
> > One follow-up comment regarding Q2: Why evaluate ground truth labels (Table 2)?: I am not sure about ground truth labels being "noise-free," as the process that generated the labels could have been noisy whereas a classifier's rules could be cleaner. I do see that it enables comparisons with human explanations.

---

### Official Review · Reviewer_BPNE · 2026-03-12

**Soundness:** 2
**Presentation:** 3
**Significance:** 2
**Originality:** 2
**Overall Recommendation:** 2
**Confidence:** 4

**Summary:**

This paper studies natural-language explanations for classifiers and argues that explanations should be evaluated primarily by behavioral faithfulness. The paper formalizes two metrics, recoverability and simulatability, and introduces OPEX, an explanation generator trained with GRPO to optimize these objectives directly rather than imitate human-written rationales. Empirically, the paper reports large gains over prompted LLaMA-3.1-8B, LLaMA-3.3-70B, and GPT-4o baselines across several tabular and text classification tasks, plus a small human study showing improved user accuracy.

**Compliance With Llm Reviewing Policy:**

Affirmed.

**Key Questions For Authors:**

Please see weaknesses above.

**Limitations:**

Please see weaknesses above.

**Strengths And Weaknesses:**

**Strengths**
1. The paper is well motivated by the goal of improving the behavioral faithfulness of explanations, particularly through the lenses of recoverability and simulatability.
2. The paper is well written and presented.

**Weaknesses*
1. The novelty of this work is moderate. The metrics are not new evaluation targets and have already been emphasized in prior work. The work MaNtLE [1] is already a natural-language explainer aimed at classifier rationale. The main novelty here seems to be replacing supervised explanation learning with RL that directly optimizes these metrics.
2. The paper needs more comprehensive baselines to verify its effectiveness. In particular, it should include a comparison with MaNtLE, which appears to be a highly relevant prior baseline. The paper should also clarify the evaluation settings for Llama-8B and Llama-70B, since it is currently unclear whether their weaker performance reflects genuine method differences or simply suboptimal prompting choices. More broadly, the current experiments do not rule out the possibility that a carefully curated few-shot prompting baseline, potentially with CoT reasoning, could achieve substantially stronger results. Without such comparisons, it is difficult to determine how much of OPEX’s advantage comes from the proposed optimization method itself versus the weakness of the chosen prompting baselines.
3. The method is evaluated only on classification tasks and this leaves it unclear whether the framework extends meaningfully to generative settings, where outputs are open-ended, behavior is multi-modal, and reward design becomes substantially more difficult. Given that GRPO is a general optimization tool rather than a classification-specific one, the motivation of only using GRPO but not formulating a
more general explanation-learning framework to generative tasks is unclear.

[1] Menon, R.R., Zaman, K., & Srivastava, S. (2023). MaNtLE: Model-agnostic Natural Language Explainer. Conference on Empirical Methods in Natural Language Processing.

---

> ### Author Rebuttal · Authors · 2026-03-30
>
> We thank Reviewer BPNE for the engagement and concrete suggestions. We agree that the paper should have included stronger baselines and more clearly positioned its novelty relative to prior work. We address each concern below.
>
> **W1: Moderate novelty—metrics known; MaNtLE exists.**
> We agree that recoverability and simulatability are not new *evaluation* notions by themselves. Our contribution is to turn them into the *training objective* for explanation generation. This changes the learning problem itself: instead of learning to imitate explanations, OPeX learns explanations *only insofar as they enable accurate prediction of model behavior*. Importantly, OPeX does not optimize explanation quality indirectly via preference or plausibility signals; it directly optimizes whether the explanation functions as a predictive surrogate, which is not captured by prior supervised or prompting-based approaches.
>
> This shift has three concrete consequences: (1) no human annotations are required, making OPeX applicable to any classifier on any domain; (2) explanation quality becomes externally testable against the model's behavior; (3) multi-task RL training enables zero-shot transfer to unseen tasks and classifiers. MaNtLE is clearly a relevant empirical comparator despite differing in supervision and granularity. The new results below show that its supervised-imitation paradigm produces explanations that are dramatically less behaviorally faithful than even zero-shot prompted LLMs.
>
> **W2: Need MaNtLE comparison; few-shot/CoT could close gap.**
> In direct response, we ran all requested baselines on the full classifier-prediction evaluation suite. OPeX-F remains substantially ahead (all pairwise differences p < 0.01):
>
> | Method | Avg Simulatability | Avg Recoverability |
> |---|---|---|
> | MaNtLE | 0.52 ± 0.22 | 0.53 ± 0.20 |
> | DeepSeek-R1-Distill-Llama-70B (CoT) | 0.66 ± 0.21 | 0.73 ± 0.21 |
> | DeepSeek-R1-Distill-Llama-70B (CoT + few-shot) | 0.65 ± 0.21 | 0.76 ± 0.23 |
> | GPT-4o (zero-shot) | 0.68 ± 0.20 | 0.73 ± 0.19 |
> | **OPeX-F** | **0.86 ± 0.13** | **0.99 ± 0.03** |
>
>
> These added baselines substantially reduce the concern that OPeX's gains come from weak prompting. MaNtLE (0.52 simulatability) falls below all prompted LLMs, CoT and CoT + few-shot perform comparably to GPT-4o zero-shot and do not close the gap, and MaNtLE is additionally not applicable to text-based tasks while OPeX handles both tabular and text domains.
>
> **W3: Only classification—unclear extension to generative settings.**
> We agree the present paper should be framed strictly as a classification-setting study, and we will narrow the framing accordingly. Classification provides the right first setting for validating behavioral-faithfulness optimization because the reward signal is unambiguous. Extension to generative settings would require redesigning reward functions and evaluation protocols—important but distinct future work.
>
> We genuinely value your rigorous review, which has substantially enhanced the clarity and robustness of our paper! Please let us know if you have additional concerns. Given these clarifications, we also invite you to reconsider your evaluation scores.

---

> > ### Author Rebuttal · Reviewer_BPNE · 2026-04-05
> >
> > Thanks for all the responses. My concerns have partially addressed, but I still maintain some questions for this work.
> >
> > Although the authors reframe the evaluation metrics as learning objectives, the core optimization method, GRPO, is itself an existing technique. As a result, the main technical contribution appears to be the application of a known RL-based optimization method to this explanation-learning setting, rather than the introduction of a fundamentally new training framework. The paper also does not clearly justify why GRPO is particularly well suited to this problem compared with other alternatives, such as DPO or other simpler post-training methods. Also, the reward signal in GRPO is produced by a User-proxy LLM, and the same overall evaluator framework is used again in automatic evaluation. This creates a risk that the method is learning to generate explanations that are especially legible to the verifier model, rather than explanations that are genuinely faithful surrogates of classifier behavior.
> >
> > The near-perfect average recoverability is striking, but it also raises concerns about whether the model is overfitting to the data itself, or whether the evaluation protocol is overly favorable to the learned explanation style.

---

> > > ### Author Response · Authors · 2026-04-05
> > >
> > > Thank you for the continued engagement. We address each point below.
> > >
> > > **GRPO is an existing technique.**
> > > We agree that GRPO is not our contribution: it is the optimization tool, not the idea. Our contribution is the formulation: casting explanation generation as an RL problem where the reward is behavioral faithfulness (simulatability/recoverability) rather than agreement with human rationales. This formulation is what eliminates the need for human annotations, makes explanation quality externally testable, and supports zero-shot transfer to held-out tasks and classifiers. GRPO is one reasonable way to optimize this objective; the contribution would likely hold under other RL methods as well.
> > >
> > > **Why GRPO over DPO or simpler alternatives.**
> > > Simpler post-training methods like supervised fine-tuning would require ground-truth explanations, which is precisely what we aim to avoid. GRPO is well suited because it constructs relative preferences from multiple sampled outputs per prompt without requiring a learned critic or pre-existing preference pairs. DPO requires paired preference data which is not naturally available, but can be similarly constructed to our setting. That said, we agree this justification should appear in the paper and will add it.
> > >
> > > **Shared evaluator concern (training reward and evaluation both use a User-proxy LLM).**
> > > This is an important concern, and one we specifically designed experiments to address. Three pieces of evidence mitigate it:
> > > - (1) Table 6 shows that switching the evaluator to GPT-4o at test time  (a different model family from the LLaMA-based training evaluator) yields nearly identical results (0.84 vs 0.86 simulatability). If OPeX were merely learning to be legible to the training evaluator, performance should degrade substantially under a different evaluator.
> > > - (2) Table 4 evaluates on synthetic tasks whose labeling rules are absent from any LLM’s pretraining data, ruling out the possibility that the evaluator is relying on prior knowledge rather than the explanation.
> > > - (3) The human user study (Figure 3) shows that OPeX explanations improve human classification accuracy by 15 percentage points: *humans are an entirely independent evaluator.*
> > >
> > > Together, these three checks make it unlikely that the gains are evaluator-specific.
> > >
> > > **Near-perfect recoverability/possible overfitting.**
> > > Recoverability measures whether the explanation describes the 25 inducing examples well enough for a reader to reproduce their labels. Near-perfect recoverability scores are not surprising, and are perhaps even expected for a good explanation: if you describe a classifier’s rules correctly, a competent reader should be able to apply those rules to the examples that motivated the description. *The more relevant test is simulatability (whether the explanation generalizes to unseen examples) and there, OPeX scores 0.86 on average, which is strong but clearly not saturated.* The gap between recoverability (0.99) and simulatability (0.86) suggests that OPeX is capturing generalizable decision patterns, not merely fitting the inducing examples. We also note that evaluation is always on held-out tasks and classifiers not seen during OPeX’s RL training.

---

### Official Review · Reviewer_aPcz · 2026-03-12

**Soundness:** 2
**Presentation:** 1
**Significance:** 2
**Originality:** 2
**Overall Recommendation:** 3
**Confidence:** 3

**Summary:**

The paper proposes a method to generate a natural language description of a machine learning model f, in order to explain the way it proceeds to make a prediction. The method, called OPEX, aims at optimizing the explanation behavioral faithfulness, ie the fact that the explanation can be used as a reliable surrogate of the model. The behavioral faithfulness is quantified through two measures, recoverability and simulatability, that assess the ability to output the same prediction as f, based on the provided explanation, respectively on the data used to train OPEX and on new data.
The paper contains an experimental study that analyses several characteristics of OPEX on 6 data sets (4 tabular ones, 2 text ones, all in a classification setting) and synthetic ones: it assesses its performance to explain ground truth labels, model predictions, it compares the results to explanations generated by three LLM (Llama 8B,
Llama 70B and GTP-4o) and to explanations written by human. It studies the impact of model size, number of samples and the final predictor (Llama 70B, GPT-o or human).

**Compliance With Llm Reviewing Policy:**

Affirmed.

**Final Justification:**

I find the core idea interesting, but I believe the paper requires deeper clarification that goes beyond what this rebuttal round can accommodate. A substantial revision would be needed for the work to meet the standards of a venue like ICML. I therefore keep my recommendation at "weak reject".

**Key Questions For Authors:**

1. Can you please clarify whether OPEX is intended to explain a given classifier, and, if so, why it is trained with examples taken from several classifiers at once?

2. In Figure 3, the accuracy reported without explanation is actually not clear: why does the value depend on the explanation generator?

3. Please clarify any point that I may have misunderstood in the strength and weakness section

Additional curiosity question: why is the OPEX variant optimized for recoverability named OPEX-F and not OPEX-R?

**Limitations:**

yes

**Strengths And Weaknesses:**

**Strenghts**

- Generating Natural Language Explanations without needing training human-written rationales, that are expensive to obtain and may not align with the actual principle of the model to be explained, is of great interest.
- The experimental study across tasks, classifiers and human user evaluation is also a strength of the paper.

**Weaknesses**

- However, as detailed below, the paper lacks clarity at several levels: the exact addressed task, the training procedure, the experimental protocol. Some of the points are made explicit only very late in the paper, whereas they should be stated already in the introduction.
- For instance, the fact that OPEX generates global explanations (as opposed to instance based, aka local, ones) is only made explicit at the end of Section 2 and the beginning of Section 3, it should be said already in the abstract and introduction.
(Also note that the subsection about instance-level and feature-based explanations in the Related works section does not seem necessary in the context of Natural Language Explanations.)
- According to the beginning of the paper, it seems that OPEX aims at explaining a given classifier, but according to the training procedure, it seems that it is trained with predictions made on different data sets by different classifiers. It is then not clear what is the aim of the proposed OPEX method: explain the behavior of one classifier or several?
- In the conducted experiments, the chosen competitors would benefit from more discussion and justification. They seem to be a very specific way of generating NLE and the exact protocol how they are used (what prompt do they answer to) is not detailed.
In the case of text classifiers, it seems the use of Chain-of-thought methods should be considered as a competitor, more discussion about their rejection seems to be needed.
- The experimental protocol does not seem to make explicit which LLM is used to make the prediction based on the OPEX explanation.
- Some formulation also seem to be clumsy: it does not seem appropriate to use the expression "OPEX accuracy" as OPEX does not perform classification. The accuracy is computed on the pair (OPEX, classifier) where the classifier is an LLM or a human user.
- From its very principle, recoverability seems to be defined as the accuracy of the pair (OPEX, classifier) on the training data, which is a metric usually discarded in favor of the accuracy on test data. Its usefulness seems debatable but is not discussed as such in the paper.
- The simulatability of (OPEX, classifier) computed on the car evaluation data set when the classifier is a human being (0.75 +/-0.15 according to Figure 3) has a high variability and is much lower than the one obtained with an LLM classifier (0.92 +/- 0.05) according to Table 2.  None of these observations is commented on in the paper.

---

> ### Author Rebuttal · Authors · 2026-03-30
>
> We thank Reviewer aPcz for the detailed feedback. We agree the paper did not make the task-level/global setup explicit early enough, and this likely caused several downstream confusions. We address each point below.
>
> **W1: Global nature of explanations should appear earlier.**
> Agreed. We will state in the abstract and introduction that OPeX produces global, task-level explanations that characterize a classifier's behavior across multiple examples, not instance-level justifications.
>
> **W2: Unclear whether OPeX explains one classifier or several.**
> OPeX explains one classifier at a time. Each input consists of samples from a single classifier on a given task, and the resulting explanation describes that classifier's decision behavior. During training, we use multiple such sets from different classifiers and tasks to learn a generalizable explanation function—but at inference time, OPeX is always applied to explain one target classifier given its examples. We will make this distinction explicit in §3.1 and §3.4.
>
> **W3: Competitors need more discussion; prompts not detailed; CoT should be considered.**
> We agree the baseline protocol was underspecified. All prompted LLM baselines use the same fixed optimal explanation prompt template (Table 8); we will surface this more prominently in the main text. We also added stronger requested baselines on all evaluation datasets (explaining classifier predictions):
>
> | Method | Avg Simulatability | Avg Recoverability |
> |---|---|---|
> | MaNtLE | 0.52 ± 0.22 | 0.53 ± 0.20 |
> | DeepSeek-R1-Distill-Llama-70B (CoT) | 0.66 ± 0.21 | 0.73 ± 0.21 |
> | DeepSeek-R1-Distill-Llama-70B (CoT + few-shot) | 0.65 ± 0.21 | 0.76 ± 0.23 |
> | GPT-4o (zero-shot) | 0.68 ± 0.20 | 0.73 ± 0.19 |
> | **OPeX-F** | **0.86 ± 0.13** | **0.99 ± 0.03** |
>
> All OPeX differences are significant (paired t-test, p < 0.01). CoT and CoT + few-shot do not improve over GPT-4o zero-shot on simulatability, and MaNtLE (supervised on human rationales) falls substantially below all prompted LLMs. These results suggest that OPeX's gains are unlikely to be explained solely by baseline weakness, and are consistent with benefits from direct RL optimization.
>
> **W4: Which LLM is used for prediction based on the explanation?**
> LLaMA-3.3-70B-Instruct serves as the User-proxy LLM in all main experiments. GPT-4o is used as an alternative evaluator (Table 6) for robustness. We will state this at the beginning of §5.
>
> **W5: "OPeX accuracy" is misleading.**
> Agreed. We will revise to "simulatability/recoverability of OPeX explanations" throughout.
>
> **W6: Recoverability is essentially training accuracy—usefulness debatable.**
> We agree our wording made recoverability sound like training accuracy. In fact, all evaluation is performed on held-out target tasks and classifiers not seen during OPeX training. Recoverability measures fidelity on the inducing examples used to generate the explanation, while simulatability measures generalization to unseen examples. We will revise §3.3 to make this distinction explicit.
>
> **W7: Human simulatability (0.75±0.15) vs LLM (0.92±0.05) not discussed.**
> The lower mean and higher variance for human evaluators reflects natural variation in attention and reasoning strategies. Importantly, OPeX still yields the largest human improvement (+15%, Figure 3). We will add this discussion.
>
> **Q: In Figure 3, why does accuracy without explanation vary across groups?**
> Baseline accuracy varies slightly (0.63–0.65) due to random participant assignment in the between-subjects design. These differences are not significant. The key is the within-group gain: +0.15 for OPeX-F vs. +0.03 and +0.00 for baselines.
>
> **Q: Why OPeX-F and not OPeX-R?**
> OPeX-F stands for "Faithfulness." We acknowledge this is confusing and are open to renaming to OPeX-R.
>
> We genuinely value your rigorous review, which has substantially enhanced the clarity and robustness of our paper! Please let us know if you have additional concerns. Given these clarifications, we also invite you to reconsider your evaluation scores.

---

> > ### Author Rebuttal · Reviewer_aPcz · 2026-04-04
> >
> > Thank you for your thorough and well-organized rebuttal. I appreciate the effort you put into addressing my concerns.
> > After carefully reviewing your responses, I have decided to maintain my original scores. While I acknowledge the clarifications you provided, they raise additional questions. If I understand correctly, OPEX is trained as a generalisable explanation function using different classifiers and applied to one target classifier: how is this application actually performed? I understand the prompt shown in Table 8 plays this role, but how does it relate  to the general explanation function trained in the first step? The authors should probably consider including a schema illustrating the whole process, showing the training step and the so-called inference step to generate the explanation of the classifier of interest. The schema (or possibly a second one) should illustrate as well the experimental protocol,  ie the combination of the generated explanation with a classifier (LLM or human) showing their application to the data used in training the classifier of interest and to new data.

---

> > > ### Author Response · Authors · 2026-04-04
> > >
> > > Thank you for your follow-up and for engaging with the paper. We would like to resolve this remaining ambiguity as clearly as possible.
> > >
> > >
> > > **How the trained OPeX model is used on a new target classifier.**
> > > OPeX is a single LLaMA-3.1-8B model with LoRA weights trained via GRPO across many (task, classifier) pairs. During training, each prompt is built from N labeled examples drawn from one single task–classifier pair; the overall training set contains many such prompts from different tasks and classifiers. The prompt template used for this is shown in Table 8.
> > > At inference time, given a new target classifier, we sample N=25 labeled examples {(x_i, y_i)} from that classifier, format them using the prompt template in Table 8, and feed that prompt to the trained OPeX model. OPeX then generates a task-level explanation of that classifier's decision behavior. *There is no additional fine-tuning or adaptation step at inference time: the general explanation capability is learned in the LoRA weights during training, while the particular classifier to be explained is specified entirely by the examples in the prompt.*
> > >
> > >
> > > **Relation to training.**
> > > During training, OPeX repeatedly sees lots of prompts of exactly this form, each built from one classifier on one task. For each prompt, it generates candidate explanations, which are scored using recoverability/simulatability via the User-proxy LLM, and GRPO updates the model weights. At inference, the same trained model is used with the same prompt format, but now on a held-out or unseen (task, classifier) pair. In that sense, *training teaches OPeX how to map a set of labeled examples from one classifier into a behaviorally faithful task-level explanation, and inference simply applies that learned mapping to a new classifier through its prompt examples.*
> > >
> > >
> > > **About the figure.**
> > > Figure 1 was intended to convey this two-phase process, but we agree that it does not currently make the separation between training and inference/evaluation clear enough. We will revise it to label these stages more clearly, indicate the role of the prompt in both, and explicitly show the downstream evaluation protocol in which a User-proxy LLM or human predicts labels using only the generated explanation. We agree that a second schematic focused specifically on the training/inference pipeline could make this substantially clearer.

---

### Decision · Program_Chairs · 2026-04-30

**Decision:**

Accept (regular)

**Comment:**

The paper studies natural language explanations for how a specific model works on an entire dataset.

The paper received mixed reviews ranging. There was a good amount of discussion between us. There were concerns about the lack of novelty and the use of GRPO. I downweighted both of these concerns due to strong empirical results and a clear presentation of the paper.

The reviewers had many suggestions that I hope the authors can incorporate into the final version. In particular, I expect the authors to add the experiments performed in response to Reviewer aPcz in the paper. Also, I think it should be made clear where the definitions of recoverability and simulatability are coming from, and those papers should be cited.